# A Randomized Controlled Trial Protocol to Test the Efficacy of a Dual-Task Multicomponent Exercise Program vs. a Simple Program on Cognitive and Fitness Performance in Elderly People

**DOI:** 10.3390/ijerph18126507

**Published:** 2021-06-16

**Authors:** Juan Antonio Párraga-Montilla, Agustín Aibar-Almazán, José Carlos Cabrera-Linares, Emilio Lozano-Aguilera, Víctor Serrano Huete, María Dolores Escarabajal Arrieta, Pedro Ángel Latorre-Román

**Affiliations:** 1Department of Musical, Plastic and Corporal Expression, Faculty of Humanities and Educational Sciences, University of Jaén, 23071 Jaén, Spain; jparraga@ujaen.es (J.A.P.-M.); platorre@ujaen.es (P.Á.L.-R.); 2Department of Health Sciences, Faculty of Health Sciences, University of Jaén, 23071 Jaén, Spain; aaibar@ujaen.es; 3Department of Statistics and Operations Research, Faculty of Social and Legal Sciences, University of Jaén, 23071 Jaén, Spain; elozano@ujaen.es; 4Faculty of Humanities and Social Sciences, International University Isabel I, 09003 Burgos, Spain; victor.serrano@ui1.es; 5Department of Psychology, Faculty of Humanities and Educational Sciences, University of Jaén, 23071 Jaén, Spain; descara@ujaen.es

**Keywords:** physical training, cognitive training, elderly people

## Abstract

Background: The necessity of improve the life quality in elderly people is well-known. The aim of this study was to determine the effects of physical and cognitive training programs, as well as their combination on the cognitive functions and physical capacities in women over 80 years old. Methods: Forty-three women took part in this study (80.86 ± 5.03 years). They were divided into four groups (three experimental groups and one control group). Experimental group 1 performed cognitive training. Experimental group 2 did physical–cognitive training, and Experimental Group 3 accomplished physical training. All of training programs had duration of eight weeks (five sessions of 60 min per week). We measured cognitive variables with the Stroop test, D2 test, and Trail Making test. Physical variables were measured with handgrip strength, Minute Step Test, and visual–acoustic reaction time. Results. Control group reduces his physical and cognitive capacities, while the three experimental groups increase these capacities. We found a strong correlation between the increase of physical and cognitive capacities. Conclusion: Eight weeks of training physical, cognitive or mixed, increased physical and cognitive functions of elderly people which may reduce the negative effects of the aging process.

## 1. Introduction

Physical activity and cognitive activity are aspects of the well-being of elderly people and are associated with health, morbidity, and mortality [1,2]. The relationship between physical and cognitive activity has been studied [3]. Recently, there has been renewed interest in physical activity associated with increased brain volume and its impact on cognitive function, which is reduced in old age as a natural aging process [4]. Therefore, it is important to focus on motor task exercises that demand attention and involve physical and cognitive capacity simultaneously (dual task). It can be a novel indicator for the detection of physical and cognitive frailty in elderly [5]. The dual-task (DT) paradigm is a relatively new method that involves the performance of two different tasks simultaneously. It was first used in psychological studies in the middle of the 1980s. The DT paradigm was then used by health professionals to assess and train elderly people as a secondary task, after walking [6].

Walking performance is considered a strong biomarker of health. In addition, walking is an automatic process in healthy adults. Nevertheless, this automatism decreases as age increase [7]. The DT paradigm detects some gait issues and cognitive deficits which are common in the elderly. This problem cannot be detected if we assess the gait of elderly people with a single task [8]. Consequently, DT can help to decrease deterioration in the elderly [9]. In addition, it can reverse or delay the onset of cognitive deficits [10], thus it is an important component of cognitive protection [11].

Physical activity has a positive effect on the cognitive reserve capacity of the brain, decreasing the effects of aging and the risk of developing neurological diseases and dementia [12]. Specifically, it can improve execution capacity and speed of information processing [13].

Increasing aerobic capacity and blood flow in the brain results in improvements in the use of oxygen and glucose in the brain, greater stimulation of neurogenesis, and an increase in synaptic interconnections [14]. Likewise, Bliss et al. [15] concluded that there is enough evidence that aerobic training can improve cognitive blood flow. Moreover, Nagamatsu et al. [16] found an association between physical fitness and cognitive performance in women aged 70–80 years and concluded that aerobic training has an influence on the improvement of verbal and spatial memory and verbal–auditory learning. Similarly, Liu-Ambrose et al. [17] found that aerobic and resistance programs enhanced selective attention, conflict resolution, and processing speed. Likewise, Vaughan et al. [18] showed that a multicomponent strength, aerobic, and motor aptitude training program had a beneficial effect on physical and cognitive performance tests as well as on peripheral blood concentrations of brain-derived neurotrophic factor (BDNF) which is a protein associated with nerve growth factor. In addition, aerobic exercise counteracts neurological decreases that are associated with cognitive factors and physical impairments [19].

Keating et al. [20] showed that a resistance training program focused on the lower body produces significant improvements in balance and gait in this population range, in addition, these authors found an increase in strength levels at this age is associated with a higher quality of life and less dependency during the aging process. In relation to a training program (aerobic or resistance), aerobic training has a positive effect on the quality of life in elderly adults since it reduces the risk of cardiovascular diseases and improves functional parameters [21].

On the whole, greater upper-body strength is considered an indicator of general vitality and a predictor of cognitive functioning, and consequently disability and mortality [1]. Taekema et al. [22] suggest that elderly adults having less grip strength correlates with lower scores in functional, psychological, and social health, predicting an accelerated decrease in the activities of daily living and cognition. In addition, a decrease in strength level correlates with an increased dependence with regard to activities of daily living [15,23], cognitive impairment [23,24], and mortality [25,26]. Reaching 65 years of age is a turning point for these capacities as there is an association between handgrip strength and cognitive performance [1]. At advanced ages, physical performance is compromised in numerous cognitive tasks that require a variety of perceptual and cognitive processes, slowing execution speed [27,28]. Therefore, the time-reaction is considered an indicator of nervous system performance that is seriously affected in elderly people [29,30].

Therefore, it is necessary to analyze the effects of the combined training programs on the physical capacities and the cognitive function of sedentary elderly people. In addition, it is important to define the criteria for prescribing a cognitive training program, a physical training program, or both. Thereby, the aim of this study was to determine the effects of physical and cognitive training programs, as well as their combination on the cognitive functions and physical capacities in women over 80 years old. Our initial hypothesis deals with the fact that a combined program of physical and cognitive exercise based on DT paradigm would have a greater effect on physical and cognitive abilities in elderly adults than when cognitive or physical training is performed independently.

## 2. Materials and Methods

### 2.1. Participants

Forty-three women (80.86 ± 5.03 years; BMI = 24.39 ± 2.44 kg/m^2^) participated in this study (Table 1). The participants attended a “day center” from Monday to Friday to carry out different activities. In Spain, a “day center” is a place which elderly people attend during a certain number of hours each day (morning or afternoon) to perform different activities directed by professionals from different fields (such as physiotherapists, psychologists, and physical trainers) aimed at improving their physical and cognitive abilities. The researchers moved to the day center to conduct the training session and to evaluate the participants (pre and post-test).

Participants were randomly assigned to the control group (CG; *n* = 11), which did not perform any training program; Experimental Group 1 (EG1; *n* = 10), which after an initial evaluation did a cognitive training program; Experimental Group 2 (EG2; *n* = 10), which combined a physical and cognitive training program; Experimental Group 3 (EG3; *n* = 12), which completed a physical training program. Randomization was conducted independently by a research assistant who was not involved in the data collection, using random numbers generated in Microsoft Excel 2010 (Redmond, WA, USA). A recruitment flowchart of the participants is reported in Figure 1.

The inclusion criteria were: (a) women older than 80 years old; (b) prosthesis-free; (c) free of any pathologies associated with an increased risk of falling (e.g., Parkinson’s disease); (e) independently ambulatory; (f) free of any disease that requires daily medications that may affect gait performance (to avoid any influence on fitness measures); (g) any pathologies or diseases that imply cardiorespiratory risk with physical effort. The exclusion criteria were: (a) cardiovascular diseases such as ischemic heart diseases or stroke; (b) any other pathology that prevents them to move without help; (c) any cognitive disease that did not allow to understand the protocol of evaluation, or the exercises to complete during training program; (e) not attending to 90% of training sessions.

We followed the ethical recommendation approved in the Declaration of Helsinki (2013). In addition, we followed the directives of the European Union on Good Clinical Practice (111/3976/88 of July 1990), as specified in a National legal framework for human clinical research (Royal Decree 561/1993 on clinical essays). The study was approved by the Ethics Committee of the University of Jaén (Protocol Code: OCT.20/7.PRY).

### 2.2. Materials and Testing

To assess physical fitness, we used different methods.

(a) 2 Minute Step Test (2MST), consisting of a gait on the site without displacement. The evaluator counted the number of times that the right knee reached the required height in 2 min. Before starting the test, we measured the height to which the participant had to raise the knee. We carried a cord from the iliac crest to the middle of the patella, then held it from the iliac crest and bent it in half, marking a point in the middle of the thigh that indicated the height of the lifted knee. To visualize the height of the step we transferred the mark of the thigh to the wall. We put a piece of tape on the wall so the participant had a visual reference. On the signal “go” the participant began stepping (not running) in place, raising each knee to the mark on the wall, for as many times as possible in the 2 min period. If the proper knee height could not be maintained, we asked the participant to slow down, or to stop until they could regain the proper form, but kept the stopwatch running. At the end of the test, we provided a cool down by asking the participant to walk slowly for a minute. A chronometer was required for this test [31].

(b) Handgrip strength (HS) was assessed in a bipedal position, with the arm stretched at an angle of 30° to the trunk, and, without support, exerting force on the dynamometer. We measured HS of the dominant hand three times. A total recovery was allowed between, and we recorded the best one for statistical analysis [32]. We used a Grip Strength Dynamometer TKK.5101 adaptive manual pressure dynamometer, with precision of 0.1 Kg, as used in previous studies [33].

(c) For measuring the visual–acoustic reaction time (V-ART) we used OptoGait system (Optogait^®^ Microgate, Bolzano, Italia) to measure V-ART. Optogait is an optical data acquisition system composed of a transmitter and a receiver bar. Each 1 m bar contains 96 Infrared LEDs (1.041 cm resolution) and is located on the transmitter bar, continuously communicating with the LEDs located on the receiver bar. The bars measure flight and contact times during execution with an accuracy of 1/1000 of a second. Optogait registered the reaction time when the participant’s foot stopped touching the ground. It showed high concurrent reliability [34]. We assessed both feet; the participant had to stand with one foot resting between the two bars of the platform and the other outside of the bars, facing a screen with speakers. The participant lifted the foot which was between the transmitter bars when a visual or auditory stimulus appeared. Two repetitions (randomized) of the visual stimulus and two of the acoustic stimuli were made per leg, selecting the average for statistical analysis.

To assess cognitive function, we used different tests:

(a) The Stroop Test [35] assessed color–word interference and selective attention. The test consisted of three subtests. In Subtest I, 10 rows × 10 columns of color names (red, blue, green, and yellow) were printed in black on white cardboard. The time needed to read the color names in a loud voice was registered. In Subtest II, the same number of correspondingly colored patches were printed (the colors had to be named). In Subtest III, the color names were printed in incongruously colored ink, e.g., the word green could be printed in red. The color of the ink was be named. We used the Spanish version [36].

(b) The D2 test [37] is a limited-time test to measure selective attention and mental concentration, through the ability to selectively attend to certain relevant aspects of a task while ignoring irrelevant ones (e.g., performing a selective search). It was completed as quickly and accurately as possible. The test consisted of 14 rows (trials), each with 47 interspersed “p” and “d” characters. The characters had one to four dashes that were configured individually or in pairs above and/or below each letter. The target symbol was a “d” with two dashes (hence “d2”), regardless of whether the dashes appeared both above the “d”, both below the “d”, or one above and one below the “d”. Thus, a “p” with one or two dashes and a “d” with more or less than two dashes were distracters. The participant’s task was to cancel out as many target symbols as possible, moving from left to right, with a time limit of 20 s/trial.

(c) The Trail Making Test (TMT) [38] consisted of two parts (TMT-A and TMT-B). The TMT-A consisted of a standardized page on which the numbers 1 to 25 were scattered within circles, and the participants were asked to connect the numbers in order as quickly as possible. Similarly, the TMT-B consisted of a standardized page that included the numbers 1 to 13 and the letters A to L, which were randomly distributed. The participants were instructed to draw lines connecting numbers and letters (ascending and alphabetical) order, alternating numbers and letters. Before starting the test, a practice trial of six items was administered to the participants to make sure that they understood both tasks. When a participant made an error during the test performance, the examiner pointed it out and explained it, then guided the participant to the last circle completed correctly. Then, the evaluator encouraged the participant to continue with the task. A maximum time of 300 s was allowed before discontinuing the test. Direct scores of TMT were the time in seconds taken to complete each task (A and B).

### 2.3. Procedures

The training program was conducted over 8 weeks and included five weekly sessions from Monday to Friday (60 min each) with a total of 40 sessions. We conducted the training over 2 weeks to familiarize the participants with the exercises and tasks that they should complete during the intervention. The sessions were applied by a specialist (physical coach or/and psychologist, in each case). The protocol for the session began with a standard warm-up (10 min), followed by the main part of the session (40 min) and then a cool down was done (10 min) for EG2 and EG3. In addition, the difficulty of the task was increased each week, with the intervention program divided in two different levels: level 1 from week 1 to 4; and level 2 from week 5 to 8 (Figure 2).

Sessions for EG1 were implemented with individual tasks. Session one was devoted to language and selective attention; session two, calculation operation; session three, to executive function; session four, to memory; and session five, to praxis and gnosis. The sequence was repeated every week. EG2 combined physical and cognitive training sessions that consisted of motor games and motor tasks with cognitive involvement (based on DT paradigm). Session modules were designed to allow the adaptation of the content to the level of the participants and to increase the difficulty each week. Intensity was adjusted to the ability to perform functional movements in which self-loading was used as resistance. We emphasized the aerobic capacity, strength, and motor aptitude. EG3 did physical training through individual tasks, based on previous studies [9]. These sessions consisted of directed physical exercises without cognitive implication, focusing on aerobic capacity, strength, and motor aptitude. Functional movements with body weight and displacements were used, adjusting the intensity every week (Figure 3).

### 2.4. Data Analysis

Statistical Analyses Data were analyzed using SPSS, v.22.0 for Windows (SPSS Inc., Chicago, IL, USA). The significance level was set at *p* < 0.05. Descriptive data are reported in terms of means, standard deviations (SD), and median (95% CI). Tests of normal distribution and homogeneity (Kolmogorov–Smirnov and Levene’s, respectively) were conducted on all data before analysis. The results of the normality and homogeneity tests revealed that some data required nonparametric tests for analysis. Consequently, we conducted Kruskal–Wallis test and Mann–Whitney *U* test. Finally, a Sperman correlation analysis between physical, and cognitive variables was done.

## 3. Results

Table 1 shows the age, and anthropometrics characteristics of the participants. No significant differences were observed in any variable.

Table 2 provides cognitive variables analyzed in the pre-and post-tests and the differences between assessments. It shows that there are significant differences among the three experimental groups, showing an improvement in each cognitive variable that we evaluated.

The results show that there are significant differences between CG and CG1, CG2, and CG3 (*p* < 0.05). It is important to emphasize that CG did not have any improvement showing a decrement in the performance in all tasks that we assessed. On the contrary, the experimental groups obtained an improvement in the performance of the cognitive variables.

Table 3 compares the results of the physical tests during the pre- and post-tests, also the increments between assessment. We found significant differences between CG and EG1, EG2, and EG3 in all variables analised. Regarding 2MST, there was significant differences between CG1 and CG3 (*p* < 0.05). In addition, we found significant differences throughout EG2 and EG3 in V-ART variable.

Table 4 reflects the Spearman correlation analysis between physical and cognitive function variables in relation to the increase in measures after the intervention (pre-test and post-test). It should be noted that there is a strong correlation among physical and cognitive variables.

## 4. Discussion

The aim of this study was to determine the effects of physical and cognitive training programs, as well as their combination, on the cognitive functions and physical fitness in women over 80 years old. Our initial hypothesis was that a combined program of physical and cognitive exercise based on a DT paradigm would have a greater effect on improved physical and cognitive abilities in older adults compared to when cognitive or physical training is performed independently.

The main finding in our study reveals that the participants responded positively to an intervention based on either cognitive or physical training or their combination in relation to the improvement of functionality in elderly people. Nonetheless, it has not been possible to corroborate our initial hypothesis since we did not find significant differences among intervention groups.

In our study, the participants who join in an experimental group (i.e., EG1, EG2, and EG3) showed a physical and cognitive improvement, independently to the training program performed. In this regard, we have not found significant different among intervention groups. We only found significant differences in some physical variables (i.e., 2MST, and V-art), the EG3 achieved greater performance. Hence, we cannot confirm which training program induces a greater improvement to the participants. Considering the results obtained by the CG, after only two months negative effects on cognitive and physical variables resulting from the aging process were analyzed. Consequently, physical–cognitive training programs applied individually or together, including DT programs, help to prevent the deterioration associated with aging. In this regard, a previous study noted that exercise can improve cognitive function and physical capacities in this population which has been related with a better quality of life [15].

### 4.1. Cognitive Variables

Regarding cognitive variables, our results showed an improvement in EG1, EG2, and in all variables that when the training program was performed in physical or cognitive capacities, or in both of them. The results of the current study agree with those found by Valencia et al. [39], which concluded that 20 sessions of combined cognitive–motor training improves selective attention and conflict resolution and interference. Moreover, Fabre et al. [40] obtained a higher incidence on cognitive function with aerobic–cognitive training than when they were conducted separately. Likewise, Vaughan et al. [18] conducted research that included a single weekly session of 60 min of multicomponent training. They obtained improvements in the Stroop Test and the Trail Making Test. 

Although, recent studies [10,40] indicated that an intervention program which included only aerobic dance exercises in women had improvements on cognitive function, being these results similar to those obtained with a cognitive training exclusively [16,40].

Therefore, aerobic training has effects on attention and execution processes in older people, both in a combined physical–cognitive program and in an exclusively physical way [15]. 

### 4.2. Fitness Variables

The Spearman correlation analysis reveals a significant association between physical capacity and cognitive functions, specifically, in concentration and executive function. Our findings confirm the association between physical capacities and cognitive capacities. It is corroborated in a previous research [3]. Consequently, physical activity through aerobic or resistance training program is a strategy to protect physical and cognitive function [10,11].

Regarding HS, the three training programs resulted in significant improvements, whereas the CG showed a decrease on it. Considering that previous studies have concluded that there is a relationship between strength levels and quality of life [1], it is necessary to keep in mind that in the elderly, physical inactivity induces a progressive loss of strength in the upper body in a short time as in the CG in our intervention. Morevover, in accordance with other authors, we found a relationship between HS and cognitive abilities [34] as a factor of cognitive protection in vulnerable ages [23], which could have an impact on better performance in health [41].

In addition, V-ART improved significantly in the three experimental groups. Specifically, the EG1, and EG2 obtained a greater improvement with respect to EG3. However, the CG got an increment in the reaction time regarding the experimental groups. Our results are contrary to the findings of previous research, in which the reaction time after physical training did not improve [29]. This could be justified by the characteristics of the implemented tasks. Our participants performed training based on DT paradigm which included decision tasks and motor-game activities whereas they have to focus their attention on another activity. It seems to be an appropriate stimulus to respond more quickly cognitively and in a motor way [42]. Herzog et al. [27] found an association among different situations that require executive coordination, such as complex video games, the commutation of tasks, and divided attention. In the current study, a test that combines visual and auditory stimuli was implemented. Both stimuli appeared randomly, which implied a complex reaction response similar to the actions that participants have to carry out in their daily life activities. In addition, the instrument used allowed discrimination between the reaction time and the movement time. Therefore, the improvement of V-ART with physical–cognitive training that simulates activities of daily living can predict a better maintenance of cognitive abilities and reduce the risk of Alzheimer’s at the end of life [27].

According to the 2MST, our result showed an improvement among the three intervention groups showing significant differences between CG1 and EG3. EG3 got a greater improvement. Nonetheless, this improvement does not appear in the CG since this group showed a decrease in relation to their pre-test data. These results are in concordance with previous studies which support the influence of aerobic capacity on the cognitive processes [16,17,27].

A limitation of this study is that the results have not been tested in relation to males. In addition, the sparce number of participants per group could be another limitation, although, the average age of the participants was over 80 years, and it was difficult to find participants to include in our study with the inclusion criteria that we mentioned above. Furthermore, the three physical tests that we conducted could be not enough to avoid a bias of sample. Future investigations must resolve these limitations. Despite these limitations, a strength of the study was the duration of our intervention since was not easy to find participants at this age who want to join in this kind of training programs.

### 4.3. Practical Application

Incorporation of training programs, physical and cognitive, is an important stimulus to improve physical capacities and cognitive functions. Specially, if the training program is based on a DT paradigm since it simulates activities of daily living [43]. Therefore, physical programs should be complemented with cognitive tasks based on motor games, individually and interacting with peers, the resolution of motor problems, and decision-making tasks based on the uncertainty of the response.

## 5. Conclusions

To conclude, performing in a physical or cognitive training, or a combination of both, over 8 weeks improves the physical capacities and cognitive functions in older people and can preserve them from the negative effects of the aging process. Consequently, lack of training at these ages (physical and/or cognitive training) can accelerate the aging process since the deterioration of functional capacities is faster when no training program is performed.

## Figures and Tables

**Figure 1 ijerph-18-06507-f001:**
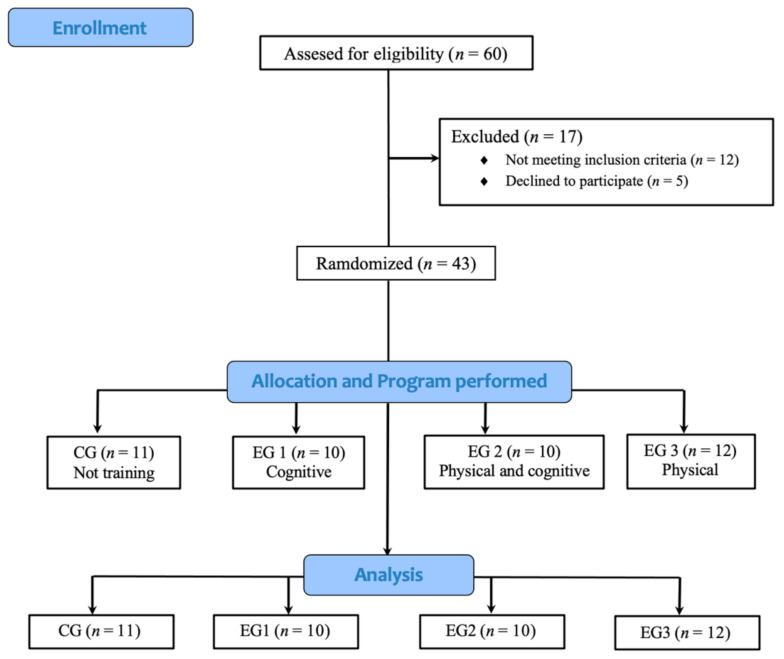
Participant’s recruitments flowchart.

**Figure 2 ijerph-18-06507-f002:**
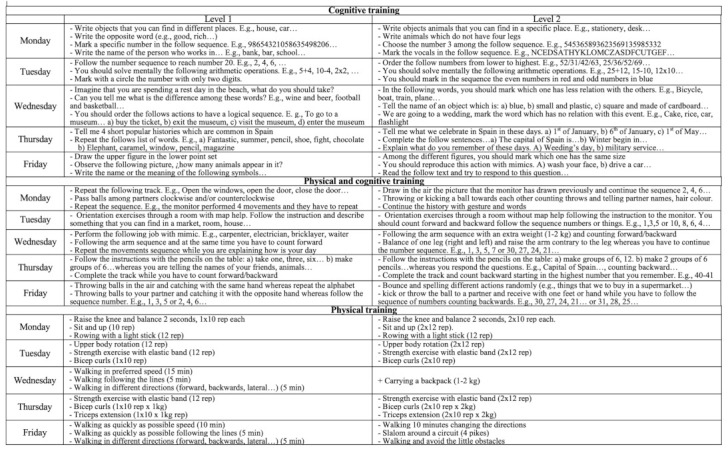
Example of the task performed by each experimental group.

**Figure 3 ijerph-18-06507-f003:**
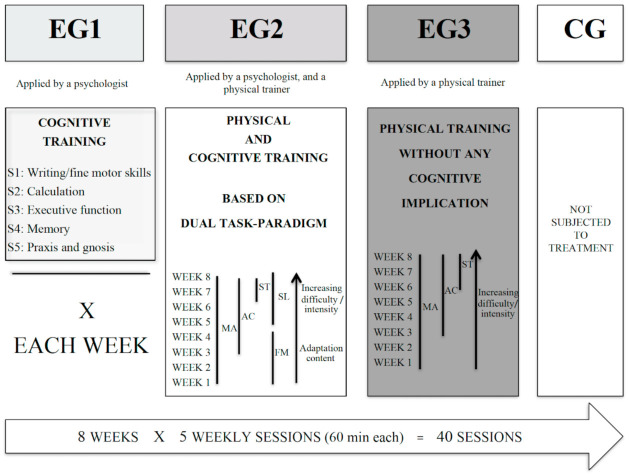
Study design. EG: experimental group; CG: control group; MA: motor aptitude; AC: aerobic capacity; S1; S2; S3; S4, and S5: sessions conducted from Monday (S1) to Friday (S5); ST: strength training; FM: functional movement; SL: self-loading.

**Table 1 ijerph-18-06507-t001:** Age, and anthropometric characteristics of the participants.

	TOTAL *n* = 43 Mean (SD)	CG *n* = 11 Mean (SD)	EG1 *n* = 10 Mean (SD)	EG2 *n* = 10 Mean (SD)	EG3 *n* = 12 Mean (SD)	*p* Value
Age (years)	80.86 (5.03)	78.54 (3.83)	82.00 (7.25)	82.70 (5.05)	80.50 (3.08)	0.240
BMI (kg/m^2^)	24.39 (2.44)	24.33 (2.09)	25.97 (3.16)	23.37 (2.24)	23.97 (1.78)	0.094
Weight (kg)	61.62 (6,51)	61.40 (5.57)	64.36 (9.25)	59.57 (5.27)	61.25 (5.49)	0.435
Height (cm)	159.00 (5.38)	158.90 (5.06)	157.40 (8.16)	159.70 (3.16)	159.83 (4.56)	0.733

SD = Standard deviation; CG: Control Group (without any training); EG1: Experimental Group 1 (cognitive training); EG2: Experimental Group 2 (cognitive and physical training); EG3: Experimental Group 3 (physical training).

**Table 2 ijerph-18-06507-t002:** Cognitive variables in pre- and post-test.

Variables	Groups	*n*	Pre-Test Median (95% CI)	Post-Test Median (95% CI)	Increase Median (95% CI)
Stroop (total points)	CG	11	13.17 (9.78/27.34)	7.57 (3.05/14.81)	−5.82 (−8.25/−4.19) a
	EG1	10	13.95 (9.65/24.87)	11.96 (8.70/19.63)	−3.23 (−4.47/−1.00) b
	EG2	10	12.90 (1.02/23.91)	9.55 (5.54/21.02)	0.14 (2.22/2.78) b
	EG3	12	17.32 (8.83/36.16)	14.71 (5.34/35.20)	−2.16 (−2.80/−0.90) b
*p*-value (group time)			0.709	0.426	0.002
D2 (total points)	CG	11	145.00 (59.00/252.00)	84.00 (69.00/136.00)	−17.00 (−94.00/2.00) a
	EG1	10	82.50 (37.01/128.00)	134.00 (68.00/166.50)	20.00 (5.00/76.50) b
	EG2	10	147.50 (96.00/229.00)	186.50 (137.00/247.00)	22.00 (9.00/42.00) b
	EG3	12	94.00 (50.00/236.00)	130.00 (48.234/122.33)	7.50 (−2.00/25.85) b
*p*-value (group time)			0.325	0.091	0.001
D2 (concentration)	CG	11	42.00 (5.00/79.00)	10.00 (7.00/15.00)	−30.00 (−68.00/5.00) a
	EG1	10	3.00 (−2.41/15.50)	20.50 (3.00/35.98)	16.00 (4.00/34.00) b
	EG2	10	32.00 (23.00/94.00)	66.00 (22.50/140.50)	10.00 (3.50/35.00) b
	EG3	12	21.00 (6.00/99.00)	22.50 (10.50/108.00)	4.50 (3.00/7.00) b
*p*-value (group time)			0.021	0.096	0.002
TMT (task A) (s)	CG	11	229.00 (122.00/349.00)	254.00 (72.89/143.39)	19.00 (−14.00/45.00) a
	EG1	10	303.50 (252.00/347.47)	211.00 (179.50/252.00)	−76.50 (−95.00/−52.00) b
	EG2	10	222.50 (146.50/291.00)	138.00 (83.00/206.00)	−66.50 (−81.00/−50.00) b
	EG3	12	205.80 (115.50/261.60)	133.20 (34.80/180.31)	−57.90 (−68.40/−48.60) b
*p*-value (group time)			0.092	0.021	<0.001
TMT (task B) (s)	CG	11	362.00 (316.00/454.00)	430.00 (361.00/470.00) a	41.00 (29.00/51.00) a
	EG1	10	415.50 (298.00/468.97)	294.50 (275.05/372.00) b	−77.00 (−116.00/−28.00) b
	EG2	10	305.00 (246.00/409.50)	275.50 (230.00/312.50) b	−59.50 (−90.00/−14.00) b
	EG3	12	324.29 (274.20/450.00)	264.00 (227.10/421.20) b	−59.39 (−69.59/−32.99) b
*p*-value (group time)			0.352	0.001	<0.001

CG: Control Group (without any training); EG1: Experimental Group 1 (cognitive training); EG2: Experimental Group 2 (cognitive and physical training); EG3: Experimental Group 3 (physical training); Stroop = Stroop Test; D2 = D2 Test; TMT: Trail Making Test; Letter in the lower case indicate a significant difference among groups; (i.e., a, b).

**Table 3 ijerph-18-06507-t003:** Physical variables in pre- and post-test.

Variables	Groups	*n*	Pre-Test Median (95% CI)	Post-Test Median (95% CI)	Increase Median (95% CI)
HS (Kg)	CG	11	17.70 (14.40/19.20)	17.00 (12.60/18.50)	−0.69 (−1.80/0.00) a
	EG1	10	12.85 (9.80/16–50)	13.25 (10.85/18.50)	1.35 (0.30/1.69) b
	EG2	10	15.95 (10.10/19.15)	17.45 (11.80/21.00)	1.65 (1.30/2.00) b
	EG3	12	15.05 (10.10/18.30)	15.95 (11.40/19.20)	1.35 (0.59/1.75) b
*p*-value (group time)			0.108	0.551	<0.001
2MST (step number)	CG	11	84.00 (77.00/86.00)	81.00 (77.00/86.00)	0.00 (−4.00/0.00) a
	EG1	10	71.50 (57.00/82.48)	78.00 (63.00/88.00)	5.50 (2.00/7.00) b
	EG2	10	83.00 (79.00/90.00)	94.50 (87.00/97.00)	11.00 (4.00/14.00) b,c
	EG3	12	82.50 (56.00/97.00)	85.50 (67.00/104.00)	9.00 (7.00/12.50) c
*p*-value (group time)			0.171	0.015	<0.001
V-ART (s)	CG	11	0.79 (0.67/1.03) a	1.03 (0.82/1.31)	0.23 (0.11/0.34) a
	EG1	10	1.26 (1.05/1.34) b	0.92 (0.85/1.05)	−0.32 (−0.37/−0.12) b,c
	EG2	10	1.18 (0.90/1.45) b,c	0.89 (0.77/1.41)	−0.33 (−0.42/−0.03) b
	EG3	12	0.89 (0.85/1.07) a,c	0.81 (0.63/0.92)	−0.20 (-0.21/-0.09)c
*p*-value (group time)			0.001	0.067	<0.001

CG: Control Group (without any training); EG1: Experimental Group 1 (cognitive training); EG2: Experimental Group 2 (cognitive and physical training); EG3: Experimental Group 3 (physical training); HS: handgrip strength; 2MST: 2 Minute Step Test; V-ART: Visual–Acoustic reaction time (second); Letter in the lower case indicate significant difference among groups (i.e., a, b, c).

**Table 4 ijerph-18-06507-t004:** Correlation analysis between physical and cognitive variables with measure increment after intervention.

	∆ Stroop	∆ D2 (Total Points)	∆ D2 (Concentration)	∆ TMT Task-A	∆ TMT Task-B	∆2MST	∆V-ART	∆HS
∆ Stroop	1	0.104	0.172	−0.352 *	−0.224	0.336 *	−0.311 *	0.219
D2 (total points)		1	0.380	−0.397 **	−0.530 **	0.511 **	−0.470 **	0.545 **
∆ D2 (concentration)			1	−0.347 *	−0.307 *	0.348 *	−0.209	0.409 **
∆ TMT Task-A				1	0.495 **	−0.350 *	0.417 **	−0.513 **
∆ TMT Task-B					1	−0.458 **	0.577 **	−0.546 **
∆ 2MST						1	−0.366 *	0.560 **
∆ V-ART							1	−0.484 **
∆ HS								1

TMT: Trail Making Test; 2MST: Minute Step Test; V-ART: Visual–Acoustic reaction time; HS: handgrip strength; * *p* < 0.05, ** *p* < 0.01.

## Data Availability

Not applicable.

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
