# Peer review of "A Randomized Controlled Trial Protocol to Test the Efficacy of a Dual-Task Multicomponent Exercise Program vs. a Simple Program on Cognitive and Fitness Performance in Elderly People"

_ijerph, 2021, doi:10.3390/ijerph18126507_

Round 1

Reviewer 1 Report

It is an interesting manuscript with original and clinically relevant data. However, the authors do not present the statistical analysis or its results correctly, so the discussions and conclusions are inappropriate.

What do you mean by "in this variable" line 82.

Figure 1 presents repetitive information. Remove the program performed section and include it within the Allocation boxes.

According to the background described, the study's main purpose was not to know if the treatments affected fitness or cognitive ability; since each treatment was designed with a different purpose or was simply different. In this sense, one hopes to see if there was a difference between the effects of each treatment (pairwise comparisons). The authors mention having performed a repeated-measures ANOVA (4x2) but do not present the results. Neither in tables presents the comparisons between treatment groups (interactions between factors by marginal measures). The correct analysis should be two-way ANOVA, treatment and time in the same analysis (4x2). Another way of knowing these effects and simplifying the analysis is to perform residual analysis (differences between final and initial: deltas) by simple ANOVA, the deltas as dependent variables and the 4 treatment groups as factors

Author Response

We really appreciate your constructive comments, helpful information and your time. Thanks to this review, our manuscript was substantially improved. Responses to your comments are written in bold. The document has been checked and we have added the information that you have suggested us.

It is an interesting manuscript with original and clinically relevant data. However, the authors do not present the statistical analysis or its results correctly, so the discussions and conclusions are inappropriate.

What do you mean by "in this variable" line 82.

We thank for your comment. We are referring to the same variable that we mention at the beginning of the paragraph which is “cognitive blood flow”. Anyway, we will change “in this variable” and we will add “cognitive blood flow” to be more specific.

Figure 1 presents repetitive information. Remove the program performed section and include it within the Allocation boxes.

We appreciate your comment. We have removed it and now it appears as you recommend us.

According to the background described, the study's main purpose was not to know if the treatments affected fitness or cognitive ability; since each treatment was designed with a different purpose or was simply different. In this sense, one hopes to see if there was a difference between the effects of each treatment (pairwise comparisons). The authors mention having performed a repeated-measures ANOVA (4x2) but do not present the results. Neither in tables presents the comparisons between treatment groups (interactions between factors by marginal measures). The correct analysis should be two-way ANOVA, treatment and time in the same analysis (4x2). Another way of knowing these effects and simplifying the analysis is to perform residual analysis (differences between final and initial: deltas) by simple ANOVA, the deltas as dependent variables and the 4 treatment groups as factors

We thank your comment. We have considered your point and after a carefully revision of statistical analysis we decided to conduct a new statistical analysis since it can explain with more accuracy the differences or not among groups. You can see it in table 2, 3, and 4.

Reviewer 2 Report

The article is interesting, but in my opinion it has many limitations:
First of all, the number of subjects in each group is very small. The authors state this in the discussion, specifically in the limitations of the research. Therefore, the results are highly biased due to the small size of the sample.

The authors should clarify sufficiently why they have not used the Senior Fitness Test by Professors Rikli and Jones. They do not clearly justify why they do not use this test, which on the other hand, is internationally recognized. This aspect should be explained more clearly in the method.

The authors' claim to assess the physical condition with only three tests is not enough, therefore they must include this aspect in the limitations. This implies modifying the bibliographic citation n 30, the protocol is not adequate, since it is applied to subjects of very different ages.

The authors say that the data tend towards normality, but I observe that the data present important standard deviations (e.g. in table 2 - D2 (total points) 129.50 (100.20)) these standard deviations indicate that the data are very disparate. Therefore, I suggest that the authors include the medians and the interquartile range in the descriptive statistics of the variables. Without this information it is not difficult for me to review the paper.
It cannot be concluded that there is improvement with these data. (e.g. 2MST) the values ​​are normal in the test for women of 80 years they are between 60-90, with which the results are within normality. I insist, it is necessary to know the medians to make an assessment of the results.

Author Response

We really appreciate your constructive comments, helpful information and your time. Thanks to this review, our manuscript was substantially improved. Responses to your comments are written in bold. The document has been checked and we have added the information that you have suggested us.

The article is interesting, but in my opinion it has many limitations:
First of all, the number of subjects in each group is very small. The authors state this in the discussion, specifically in the limitations of the research. Therefore, the results are highly biased due to the small size of the sample.

We thank your comment. You are right, we have a small sample in each group and it could be a bias. Nevertheless, previous authors have conducted another randomized control trials with similar sample size, also with dual-task [1–4]. That is the reason why we though that our sample was adequate to conduct our intervention. Also, we have added that the importance to carry out future intervention with a greater number of participants to avoid the bias of sample that you told us.

  1. Beauchet, O.; Dubost, V.; Aminian, K.; Gonthier, R.; Kressig, R.W. Dual-task-related gait changes in the elderly: Does the type of cognitive task matter? J. Mot. Behav. 2005, 37, 259–264.
  2. Brachman, A.; Marszałek, W.; Kamieniarz, A.; Michalska, J.; Pawłowski, M.; Akbaş, A.; Juras, G. The effects of exergaming training on balance in healthy elderly women—a pilot study. Int. J. Environ. Res. Public Health 2021, 18, 1–12, doi:10.3390/ijerph18041412.
  3. Tasvuran Horata, E.; Cetin, S.Y.; Erel, S. Effects of individual progressive single- and dual-task training on gait and cognition among older healthy adults: a randomized-controlled comparison study. Eur. Geriatr. Med. 2021, 12, 363–370, doi:10.1007/s41999-020-00429-5.
  4. Kim, S.J.; Yoo, G.E. Rhythm-Motor Dual Task Intervention for Fall Prevention in Healthy Older Adults. Front. Psychol. 2020, 10, 1–8, doi:10.3389/fpsyg.2019.03027.

The authors should clarify sufficiently why they have not used the Senior Fitness Test by Professors Rikli and Jones. They do not clearly justify why they do not use this test, which on the other hand, is internationally recognized. This aspect should be explained more clearly in the method.

Thank you for your comment. You are right, the Senior Fitness is widely used to measure physical condition in the elderly population. Nonetheless, we decided to use a specific test due to the age of the participant was high also we considered the physical characteristic of the participants. We wanted to avoid any risk of injury or abandonment, caused by the inability to complete the physical battery test due to difficulty in performing some of these tests and the number of tests to be performed. That is the reason why we selected different test that allowed us to obtain a good quality of data and collect information that responds to the objective of our research. Notice that, it have been reported in previous studies that at this ages some of the participants had to be removed to the intervention group due to the impossibility to get an assessment good enough to include in statistical analysis [5].

  1. Bischoff, L.L.; Cordes, T.; Meixner, C.; Schoene, D.; Voelcker-Rehage, C.; Wollesen, B. Can cognitive-motor training improve physical functioning and psychosocial wellbeing in nursing home residents? A randomized controlled feasibility study as part of the PROCARE project. Aging Clin. Exp. Res. 2020, 943–956, doi:10.1007/s40520-020-01615-y.

The authors' claim to assess the physical condition with only three tests is not enough, therefore they must include this aspect in the limitations. This implies modifying the bibliographic citation n 30, the protocol is not adequate, since it is applied to subjects of very different ages.

We appreciate your comment. The reason to include only three tests to assess physical condition was because we wanted to focus in a specific test that allow us to report an important information about the physical condition of the participants in order to avoid any risk of injury or dropout. That is why we only choose handgrip strength, 2 minutes step test and V-ART test. Anyway, we will add this information in limitations section as you recommend us.

Regarding to the protocol that we use to measure handgrip strength we have removed the citation n30 as you told us since the information is not clear so we apologize for this mistake. We wanted to express some information about the reliability to the instrument that we used (Grip Strength Dynamometer TKK.5101) which has been use in previous research [6]. In addition, we have rewritten the paragraph and now is clearer than before.

  1. Pedrero-Chamizo, R.; Albers, U.; Tobaruela, J.L.; Meléndez, A.; Castillo, M.J.; González-Gross, M. Physical strength is associated with Mini-Mental State Examination scores in Spanish institutionalized elderly. Geriatr. Gerontol. Int. 2013, 13, 1026–1034, doi:10.1111/ggi.12050.

The authors say that the data tend towards normality, but I observe that the data present important standard deviations (e.g. in table 2 - D2 (total points) 129.50 (100.20)) these standard deviations indicate that the data are very disparate. Therefore, I suggest that the authors include the medians and the interquartile range in the descriptive statistics of the variables. Without this information it is not difficult for me to review the paper.
It cannot be concluded that there is improvement with these data. (e.g. 2MST) the values ​​are normal in the test for women of 80 years they are between 60-90, with which the results are within normality. I insist, it is necessary to know the medians to make an assessment of the results.

We appreciate your point. We have added the median and the interquartile range data as you can see in table 2 and 3. In addition, we have conduct the statistical analysis again to make sure that our results has a greater accuracy than before.

Reviewer 3 Report

Dear authors,

i think your article is interest and can be good but it needs serious
improvements:

1. In abstract you wrote ages instead of years in methods. You repeat
the information about experimental group 3 in methods. Change.
2. Introduction must have more information about the importance of dual task in older adults.
3. Methods: Include exclusion criteria ; Explain with more detail what every group performed during the training program. With your information its not possible to be replicated by other reader. Figure 2 must be before data analysis.

  1. Your results are confussing. You must improve the presentation of
    table 2 and 3. Example: there is no “n” in none of the groups.
    5. Discussion must be improved. You can not only discuss your results
    on the basis  that your results are similar or different from other
    studies. You have to debate them like you do on paragraph 324-336. You
    have to discuss your results between your groups.

    I strongly recommend you to work on your article to be considered for
    publication.

Author Response

We really appreciate your constructive comments, helpful information and your time. Thanks to this review, our manuscript was substantially improved. Responses to your comments are written in bold. The document has been checked and we have added the information that you have suggested us.

Dear authors,
i think your article is interest and can be good but it needs serious
improvements:
1. In abstract you wrote ages instead of years in methods. You repeat the information about experimental group 3 in methods. Change.

We apologize for this mistake. We have changed it.
2. Introduction must have more information about the importance of dual task in older adults.

We thank your comment. We have added more information about Dual Task Paradigm as you recommend us.

The dual-task (DT) paradigm is a relatively new method that involves the performance of two different tasks simultaneously. It started to use in the middle of the 80s and this method was used by psychological studies at first. Afterward, health professionals used dual-task to assess and train elderly people as a secondary task, being walking the main task to perform [1]. DT paradigm allows detecting some gait issues and cognitive deficits which are common in the elderly as a consequence of the aging process. Noticed, that this problem cannot be detected if we assess the gait elderly with a single task [2].

  1. Woollacott, M.; Shumway-Cook, A. Attention and the control of posture and gait: A review of an emerging area of research. Gait Posture 2002, 16, 1–14, doi:10.1016/S0966-6362(01)00156-4.
  2. Bridenbaugh, S.A.; Kressig, R.W. Laboratory review: The role of gfait analysis in seniors’ mobility and fall prevention. Gerontology 2011, 57, 256–264, doi:10.1159/000322194.
  3. Methods: Include exclusion criteria ; Explain with more detail what every group performed during the training program. With your information its not possible to be replicated by other reader. Figure 2 must be before data analysis.

We appreciate your comment. We have added the exclusion criteria. Also, we have added more information about the training program as you can see in table 2. In addition, we have moved the figure 2 before data analysis as you recommend us.

  1. Your results are confussing. You must improve the presentation of
    table 2 and 3. Example: there is no “n” in none of the groups.

We thank your comment. We have added “n” in table in 2 and 3.
5. Discussion must be improved. You can not only discuss your results
on the basis  that your results are similar or different from other
studies. You have to debate them like you do on paragraph 324-336. You
have to discuss your results between your groups.

We appreciate your comment. We have added more information in discussion section as you recommend us.
I strongly recommend you to work on your article to be considered for
publication.

Round 2

Reviewer 1 Report

The authors have done a good job.

Just two comments.

1) Figure 3 presents design errors.

2) Put at the bottom of Tables 2 and 3 which groups correspond to subscripts a, b and c mentioned.

Author Response

Dear sir,

We appreciate your comment. According to your question, we will try to respond to it.

1) Figure 3 presents design errors.

We have added more information, also we have changed some design errors as you told us. Please, if you consider that the errors continue feel free to indicate exactly where it is.

2) Put at the bottom of Tables 2 and 3 which groups correspond to subscripts a, b, and c mentioned.

     Thank you for your comment. We will try to explain to you what exactly the letters mean since it is not associated to any specific group. 

     Differents letters mean significant differences between groups. However, if there is the same letter it means that there are no significant differences between groups. E.g., In table 2, appears to letter a, and b so it means that there are significant differences among CG (a) and CG1 (b), CG2 (b), and CG3 (b), nevertheless there are no significant differences among CG1, CG2, and CG3.

     We hope that now is clearer than before, if not, feel free to ask again. 

     Thanks in advance to help us to improve our paper.

Reviewer 2 Report

The research has important limitations due to the characteristics of the sample, but the statistical estimates are relatively solid, so the paper could be published. 

Author Response

    We appreciate your comment.

    Our paper has improved a lot since the changes that you recommend to us have been so useful.

   Thank you so much.

Reviewer 3 Report

Dear authors,

congratulations for your improvements.

Best regards

Author Response

We appreciate your comment.

   We think that our paper has been improved since the commentaries that you mentioned have been so useful.

    Best regards,